# Student Engagement in Patient Safety and Healthcare Quality Improvement: A Brief Educational Approach

**DOI:** 10.3390/healthcare12161617

**Published:** 2024-08-14

**Authors:** Ileana Chavez-Maisterra, Ana Cecilia Corona-Pantoja, Luz Elena Madrigal-Gómez, Edgar Oswaldo Zamora-González, Luz Berenice López-Hernández

**Affiliations:** 1Departamento Académico de Ciencias Clínicas, Universidad Autónoma de Guadalajara, Av Patria 1201, Zapopan 45129, Jalisco, Mexico; ileana.chavez@edu.uag.mx (I.C.-M.); ana.ceciliacorona@edu.uag.mx (A.C.C.-P.); luz.madrigal@edu.uag.mx (L.E.M.-G.); edgar.zamora8148@academicos.udg.mx (E.O.Z.-G.); 2Departamento de Bienestar y Desarrollo Sustentable, División de Cultura y Sociedad, Centro Universitario del Norte, Universidad de Guadalajara, Colotlán 46200, Jalisco, Mexico

**Keywords:** patient safety, healthcare quality improvement, medical students, engagement tools, healthcare education

## Abstract

Achieving optimal patient safety (PS) remains a challenge in healthcare. Effective educational methods are critical for improving PS. Innovative teaching tools, like case-based learning, augmented reality, and active learning, can help students better understand and apply PS and healthcare quality improvement (HQI) principles. This study aimed to assess activities and tools implemented to improve PS and HQI education, as well as student engagement, in medical schools. We designed a two-week course for fourth-year medical students at the Autonomous University of Guadalajara, incorporating Fink’s taxonomy of significant learning to create engaging activities. The course featured daily synchronous and asynchronous learning, with reinforcement activities using tools, like augmented reality and artificial intelligence. A total of 394 students participated, with their performance in activities and final exam outcomes analyzed using non-parametric tests. Students who passed the final exam scored higher in activities focused on application and reasoning (*p* = 0.02 and *p* = 0.018, respectively). Activity 7B, involving problem-solving and decision-making, was perceived as the most impactful. Activity 8A, a case-based learning exercise on incident reporting, received the highest score for perception of exam preparation. This study demonstrates innovative teaching methods and technology to enhance student understanding of PS and HQI, contributing to improved care quality and patient safety. Further research on the long-term impact is needed.

## 1. Introduction

Achieving optimal patient safety (PS) continues to be a challenge in healthcare delivery. The World Health Organization (WHO) estimates that around 1 in every 10 patients is harmed in health care and that more than 3 million deaths annually occur due to unsafe care, with above 50% of adverse events leading to harm being preventable [1]. Improving PS and quality in health systems requires healthcare professionals and students to actively learn from errors and to be able to apply knowledge on PS, evidence-based medicine, and healthcare quality improvement (HQI) learned during undergraduate level training [2]. Effective educational models are critical to achieving this goal. Because of this, the WHO published a patient safety curriculum guide that covers the following eleven critical topics [3]: (1) Defining patient safety, (2) human factors engineering, (3) understanding complex systems, (4) effective teamwork, (5) learning from mistakes, (6) managing clinical risk, (7) quality improvement methods, (8) patient and caregiver interactions, (9) infection control, (10) safety in invasive procedures, and (11) medication safety. This guide has been adopted for PS and HQI courses by various universities with different approaches [4,5,6,7,8]. Common educational approaches in teaching PS include concept descriptions, cause/systems analysis, communication, adverse event reporting skills, teamwork, and quality improvement principles [9]. Evidence supports the effectiveness of PS education interventions, such as a study demonstrating significant improvements in medical students’ attitudes towards PS lasting up to one-year post-intervention [10]. The landscape of medical education is evolving to incorporate cutting-edge tools, like artificial intelligence, virtual/augmented reality (AR), and gamification [11,12,13]. Innovative teaching methods, like escape room-based simulations [5,14], teaching the PDSA (plan, do, study, act) cycle using a Mr. Potato Head team-building exercise [4], as well as interactive online platforms, like “Safety Quest” [15], engage students and reinforce their knowledge, skills, and confidence in applying PS and HQI principles.

However, a significant gap exists between the theoretical knowledge gained during education and its practical application in clinical settings [2]. This is evidenced by research showing that only 19.9% of undergraduate health sciences students demonstrate good patient safety practices [16]. Newly graduated nurses, for example, may struggle to translate academic knowledge to real-world situations, potentially compromising patient safety [17]. Bridging this gap requires a deeper understanding of healthcare professionals’ knowledge needs. Previous studies have shown that undergraduate students who have received training in PS principles consider its inclusion in the curriculum to be of great importance and seem to better recognize how adopting a PS culture enhances the proper execution of medical procedures and patient care [6,18,19].

Effective instructional design plays a crucial role in leveraging new technologies to create engaging learning experiences that foster the development of essential competencies in healthcare professionals [20]. In recent years, enhancing student engagement has become a key objective in education due to its direct correlation with academic success, increased graduation rates, and overall improved well-being [21]. In this study, we adopted a comprehensive four-dimensional framework to maximize student engagement in learning activities, encompassing behavioral, emotional, cognitive, and proactive engagement. Behavioral engagement involves active student participation and adherence to classroom norms. Emotional engagement focuses on fostering genuine interest, enjoyment, and a sense of belonging. Cognitive engagement involves the use of advanced learning strategies, while proactive engagement encourages students to take the initiative in enhancing their learning experiences [21]. Miller’s Pyramid of Clinical Competence is a framework for assessing clinical skills in medical education [22]. It consists of four levels: knows, knows how, shows how, and does. This model helps educators evaluate and structure the development of clinical competencies in medical students [22,23]. While Miller’s pyramid focuses on what students should be able to do, we considered Fink’s taxonomy for the instructional design of the course. Fink’s taxonomy of significant learning provides a well-established framework for educators to design meaningful and engaging educational activities that promote deep understanding and lasting retention [24,25]. This learner-centered design extends beyond traditional cognitive domains, offering a cyclical model that promotes deeper emotional engagement and ultimately guides students towards self-directed learning [26]. It is characterized by six interactive dimensions that guide the development of learning activities, as follows (Figure 1):Foundational knowledge: comprehending and retaining core concepts.Application: utilizing knowledge to solve problems and make practical decisions.Integration: connecting information and ideas across disciplines and experiences.Human dimension: understanding the human aspects of a situation, including ethics and values.Caring: developing empathy and compassion for others.Learning how to learn: acquiring skills for independent and lifelong learning.

An initial course on PS and HQI was meticulously designed and implemented as part of the mandatory curriculum of fourth-year medical students at the Autonomous University of Guadalajara for the 2020–2021 academic year. However, after reviewing the results and students’ perceptions, the aforementioned course was modified and new reinforcement activities were introduced, with the objective of bridging the gap between theoretical content and practical applications, as well as encouraging the reporting of adverse events as a preventive measure [27].

In this context, we proposed a novel educational approach using Fink’s taxonomy as a framework to bridge the gap between theoretical knowledge and clinical practice in HQI and PS. Implementing Fink’s framework encourages instructors to consider all six dimensions, enabling the inclusion of activities that might be overlooked by other frameworks. This creates a richer learning environment that prepares students for cognitive challenges as well as the emotional and ethical complexities that they will face in patient management. This article focuses on tools and activities that facilitate the education of healthcare personnel in PS and HQI principles. Educational tools for health sciences students should offer a flexible learning experience that explores human factors, system elements, and environmental interferences contributing to incidents of patient harm. The aim of this study is to share insights gained from implementing classroom activities and tools in a seminar specifically designed for final-year medical students.

## 2. Materials and Methods

### 2.1. Course

The course was mandatory and designed to be completed within the two-week timeframe allotted in our university’s medical school curriculum, featuring two hours of daily synchronous face-to-face teaching and an additional hour designated for online (asynchronous) independent content review. The two-hour synchronous instruction was divided into two segments: the lecture hour (the first hour focused on comprehensive lectures covering essential topics) and the reinforcement hour (the second hour emphasized interactive learning reinforcement activities to consolidate knowledge). The course also provided insights into national and international hospital accreditation/certification requirements. It covered the Mexican General Health Council’s “Essential Actions for Patient Safety” agreement. A detailed syllabus outlined the course policies, required texts, schedule, and grading criteria. The grading was structured as follows: daily voluntary participation accounted for 10%, the final exam constituted 20%, teamwork on the final project made up 10%, and learning reinforcement activities comprised the remaining 60% [27].

### 2.2. Modified Reinforcement Activities

New reinforcement activities were designed and implemented in the 2023–2024 academic year by A.C.C.-P. and L.B.L.-H., both of whom received training in pedagogy and artificial intelligence in higher education from Cintana Education [28]. Seven types of activities were introduced to the course using tools, like learning apps, H5P, patient safety AR, and artificial intelligence. Each activity was graded using a rubric, quiz, or an online questionnaire (Figure 2). The goal of these modifications was to increase student engagement, skill development, and to change attitudes towards error reporting and learning from mistakes. The course comprised a total of nine activities, each divided into two parts, A and B. Two activities were designed to address challenges identified in a previous study [27]: (1) a negative student attitude toward reporting errors, and (2) the limitations imposed by the pandemic on in-class augmented reality group activities. These activities are analyzed separately due to their distinct origins.

#### 2.2.1. Assessing Student Competency in Patient Safety Reporting

This new activity directly addressed student resistance to reporting adverse events, a previously identified issue. Thus, activity 8A evaluated whether students were able to identify, analyze, classify, and report an adverse event, based on the work of Mohsin et al. [29]. Students watched a video depicting an adverse event and then applied their knowledge to classify and report it using a standardized format, identify contributing factors, and select the most relevant International Patient Safety Goal (IPSG) to prevent similar incidents. This exercise assessed their understanding of adverse event classification, reporting procedures, and the application of specific safety goals to real-world scenarios.

#### 2.2.2. Student Perceptions of a Patient Safety AR Learning

For activity 8B, groups of five people downloaded the AR application and conducted a simulation of four hospital rooms programmed in the simulator, namely the emergency department and three other hospital rooms. In these rooms, they identified patient risks, marked them, and continued with the simulation [30]. Afterwards, they completed a questionnaire about their experience using the augmented reality simulator, recalling the dangers they detected in the different rooms.

### 2.3. Final Exam

The final exam was a standardized online multiple-choice test taken on campus, consisting of 33 items. Each question had five answer options, with only one correct answer. Each correct answer was worth 3.33 points, making the total possible score 109.89 points if all questions were answered correctly. The passing threshold was set at 60 points. The exam covered a wide range of course topics, including reasoning skills, definition comprehension, and image analysis. Students had one hour to complete the assessment. The questions were drawn from a question bank created by the course professors, and a dichotomous Rasch analysis using Jamovi [31] was conducted to assess participant performance and item difficulty. The reliability coefficient was determined to be 0.72, indicating a satisfactory level of consistency.

### 2.4. Participants and Procedures

This pilot study included 394 fourth-year medical students from the Autonomous University of Guadalajara (UAG), a private university in Mexico, enrolled in the course during the 2023–2024 academic year. Students were invited to participate in this educational research project on a voluntary basis, both verbally and through an online questionnaire. To ensure honest and unbiased responses, we emphasized that participation would not affect their grades and that their responses would remain anonymous. Four professors, members of the Patient Safety and Healthcare Quality Academy at the same institution, also contributed to this study by teaching the course and distributing the questionnaire. Non-probability sampling was employed, and only duplicate questionnaire responses were excluded. The modified course activities were planned at the end of the 2022–2023 academic year and implemented from August 2023 to May 2024. The two-week course was conducted cyclically, with different groups of students participating throughout the academic year, during which time the data were collected. Statistical analysis of the data was carried out in June 2024.

### 2.5. Outcomes and Instruments

To evaluate the experience with the implementation of activities, three main outcomes were considered: the performance in the activities and their influence on exam results, the perception of the course before and after, and the perceived difficulty and usefulness of each activity at the end of the course.

#### 2.5.1. Exploring the Relationship between Performance in Course Activities and Exam Outcomes

To investigate potential associations between performance in specific course activities and final exam outcomes, Mann–Whitney U tests were employed. These non-parametric tests compared the scores on each activity between students who passed the exam and those who did not pass, as well as the influence of sex in test and activities’ scores. Students who did not complete a particular activity were excluded from the Mann–Whitney U test for that activity. The statistical analysis was carried out using JASP (Version 0.18.3) [32].

#### 2.5.2. Perceptions of Quality and Patient Safety

Two questions were posed to the students, one at the beginning and one at the end of the course, respectively: “How did I begin the course?” and “How do I leave the course?”. The responses were classified into positive and negative categories to obtain a clear view of the course’s impact. The data obtained were analyzed through a McNemar test (response rate of 12%).

#### 2.5.3. Student Satisfaction and Perceptions of Course Activities

To assess student satisfaction and perceptions of the implemented activities, an online questionnaire was administered at the course conclusion. The overall satisfaction with the course activities was evaluated using a 5-point Likert scale. Also, students rated each activity from 0 to 100 based on how much they felt it helped them prepare for the exam. Finally, the activities were ranked first by perceived difficulty and second by their impact on the student’s learning experience. However, the loss of contact with some students after the course ended and the voluntary nature of the survey resulted in limited data availability, with a response rate of 11%.

### 2.6. Data Analysis

Statistical analysis of the data was conducted using IBM SPSS version 25. A Kolmogorov–Smirnov test was employed to assess the normality of the data. Descriptive statistics provided the mean, minimum, and maximum scores for each activity. The Mann–Whitney U test was utilized for comparisons between two groups, such as male and female students or those who passed versus those who failed the final exam, with respect to their performance in each activity and the final exam. The chi-squared test was used to determine the association between sex and passing the final exam. A McNemar test was performed to evaluate changes in students’ perceptions of the course.

### 2.7. Ethical Approval

This study was registered and approved by our institutional research coordination. Participation was voluntary and the answers from individuals were anonymized for research purposes. The number assigned to this project by local committees was 20-130704.

## 3. Results

The present work shows the results of implementing classroom activities and tools in a seminar focused on PS and HQI, specifically designed for final-year medical students.

Each activity in the course had a different total possible score based on its difficulty level. Since not all students completed every activity, only those who did were included in each activity’s analysis; therefore, total participants in each activity varies (Table 1).

### 3.1. Relationship between Performance in Course Activities and Exam Outcomes

The students who passed the exam achieved higher average scores in activities 2A and 7B compared to those who failed (*p* = 0.02 and *p* = 0.018, respectively). The Mann–Whitney U test revealed that female students generally achieved better test results in the exam (*p* = 0.032, z = −2.145, U = 15859.5, Md = 76.67 and 80, men and women, respectively) (Figure 3); however, the Chi-squared test, after Fisher’s correction, showed that sex was not a variable significantly associated with passing or failing the final exam (*p* = 0.053).

Activity 7B engaged students in reasoning, hypothesis testing, problem-solving, and decision-making. Students analyzed written scenarios in which quality improvement actions were taken and the results should be analyzed, differentiating qualitative and quantitative data, determining sample types, selecting appropriate statistical tests, and interpreting *p*-values calculated through online tools. They then completed a platform quiz to reinforce their understanding. Interestingly, in the post-course satisfaction and perceptions questionnaire, most students agreed that activity 7B was both the most challenging and the most impactful on their learning experience during the course.

Activity 2A evaluated students’ vocabulary, concerning PS and HQI, and cognitive skills through a crossword puzzle. Participants who scored higher on the crossword activity tended to perform better on the exam.

### 3.2. Incident Reporting Activity

Activity 8A focused on recognizing and reporting medical errors through a case-based learning exercise. In the post-course questionnaire, students perceived the incident report activity as the most helpful for exam preparation, giving it a score of 85 out of 100. In five out of the six questions that comprised the activity, most students provided correct answers. However, for the question about contributing factors to the adverse event, less than 20% of students answered correctly (Table 2).

### 3.3. Augmented Reality Simulation Activity

Regarding activity 8B, to gain insights into students’ experiences using the Patient Safety AR application as an educational tool in the classroom, a brief questionnaire was administered after completing the simulation (Table 3).

### 3.4. Satisfaction and Perceptions of Quality and Patient Safety Course and Implemented Activities

A pre–post analysis of 51 students’ responses using the McNemar test revealed a highly significant positive change in their perceptions towards the quality and patient safety course (*p* < 0.0001). Additionally, in the post-course satisfaction and perceptions questionnaire, 80% of students expressed satisfaction with the course activities (Figure 4). This suggests that the learning experience in this course fosters a more favorable student perspective on these crucial healthcare topics.

## 4. Discussion

Herein, we addressed the challenge of teaching medical students about PS and HQI with an emphasis on reducing the gap between theoretical and practical knowledge through active learning-based activities within a tight two-week timeframe using carefully designed activities for maximum impact. Medical errors that detract from quality care are diverse (carelessness, ignorance, etc.) and can occur at any stage of care, impacting all medical specialties. The core challenge lies in preparing students with theoretical knowledge and practical skills for preventing and addressing medical errors across various clinical scenarios. This study aimed to create a meaningful learning experience.

Overall, most students perceived the activities positively, despite the challenge of evaluating long-term learning outcomes in a two-week course. The majority of activities contributed to comprehensive learning for final-year medical students. However, our data suggest that additional reinforcement is needed for students to accurately distinguish contributing factors in adverse events. Despite a correct answer rate of over 79% for most questions in the Incident Report on Patient Safety activity, students struggled to accurately identify contributing factors to adverse events. This contrasts with the findings by Mohsin et al., where students successfully identified multiple contributing factors to reported adverse events. However, in Mohsin et al.’s study, these real-life events were witnessed during hospital rotations, and reviewers had no controlled way of verifying if all contributing factors were accurately reported [29]. In the aforementioned study, a 4 h workshop was conducted, which included the viewing of a medical drama episode followed by an overview of error analysis applied to events depicted in the show and, finally, a discussion on conditions contributing to errors, types of errors, error prevention, and interventions. This proved to be effective in improving the students’ capacity to correctly identify, categorize, and report errors and their contributing factors. Since the timeframe and total number of topics covered in our course were not compatible with this exercise, a revised and modified version could be implemented in the future.

During the AR Patient Safety Simulator activity, 63% of participants reported encountering difficulties (Table 3). However, these issues were primarily technical rather than related to understanding the material. The AR app was often incompatible with their devices or failed to load correctly. Despite these challenges, most students were satisfied with the simulator experience and perceived it to be applicable to real clinical settings.

In our study, we found that female students had significantly higher test scores in the final exam than male students. However, gender was not significantly related to better performance in activities, suggesting that a good performance in activities and being female are two independent factors associated with better exam results. A study by Wu et al. on medical students found that, while male students reported significantly higher intrinsic motivation, female students had higher levels of academic performance. They attributed this difference to personality traits displayed by females in medical school, such as helpfulness and relationship consciousness, which could be valuable for success in assignments and exams. They also noted that female students might be more concerned with proving themselves academically [33]. Other studies have also found that academic performance is significantly relates to gender, with female students performing better [34,35]. Interestingly, these findings contrast with those of several other studies, which found no difference in academic performance by gender [36,37,38]. One study even found that male students showed more interest in patient safety education [39]. Although more research is mandatory, our study’s findings regarding test performance could be explained by a phenomenon in patient safety topics, where certain personality characteristics, such as empathy, people skills, and the ability to form human connections, could be more prominent in female students. In one of the aforementioned studies, the “Interest-motivated” profile, interpreted as a person studying medicine only due to their interest in patients or biology, had the highest percentage of female students [35]. It could also be that female students perform better in exams across all subjects at our university. As previously stated, further studies are needed to understand this gender-based difference.

While the final exam may not have captured the full impact of all activities, additional approaches likely contributed to student development. In this case, activities that imply reasoning and hypothesis testing had the highest impact in exam performance. Real-life videos were employed to highlight the human cost of medical errors and cultivate empathy and a patient-centered perspective. Gamified online activities aimed to combat boredom while reinforcing classroom concepts. Case analysis with a focus on error identification and root cause analysis demonstrated the effectiveness of active learning in fostering practical skills. Student-created podcasts showcased their understanding of patient safety terminology. The integration of artificial intelligence (AI) enhanced the learning experience by allowing students to create clinical risk plans and surgical adverse event infographics, equipping them to tackle complex healthcare challenges. Engaging activities, like the augmented reality patient safety simulator and podcast creation activity, motivated students and encouraged critical thinking and problem-solving. These activities provided a creative platform for students to effectively apply and solidify their knowledge.

PS outcomes most relevant to primary care include enhancing the patient experience, reducing medical errors, improving operational efficiency, and contributing to better population health. Despite its prevalence, ambulatory care lacks dedicated patient safety research. Notably, a study of ten patient-centered medical homes revealed that patients identified additional safety strategies beyond those recognized by clinicians, highlighting the need for patient-centered safety initiatives and early PS education [40].

In Mexico, general medicine doctors make up 67% of the physician workforce, dominating primary care [41]. Primary care services in the National Health System cater to approximately 85% of the population [42]. As fourth-year medical students prepare to begin their clerkship year and are on the cusp of graduation and entering the field as general medicine doctors, our educational intervention is ideally timed to equip them with the knowledge they need to apply in real-life practice and ultimately improve healthcare outcomes.

Various studies have observed that medical errors made by health students during their practice are commonly attributed to lack of experience or knowledge, insufficient supervision, fatigue from long hours, and fear of criticism, leading to underreporting of their errors [43,44,45]. To reduce the incidence of these errors and improve PS, studies suggest implementing specific educational strategies. Simulation activities, like the one included in our course, allow students to practice medical procedures in a controlled, risk-free environment, significantly improving their competence and confidence [43,44]. It has been shown that implementing a PS and HQI curriculum at any level of medical education or even during residency has a positive impact on knowledge and confidence in those topics, which in the long run helps create a more robust safety culture and promotes the development of PS-centered projects. This is beneficial for patients and enhances primary care quality [46].

A study on the perception of PS in a primary care clinic revealed a low number of reported events, with 66.7% of the nursing staff not having reported any events, and 14.8% having reported only 1–2 events in the last year. This contrasts with the perception of the patient safety culture, which 44% of the staff identified as deficient or bad [47]. Similarly, a survey of medical students during their social service year revealed a low rate of adverse event reporting; approximately 44% had never reported an event, and 35.7% reported only one to two incidents. Fear of repercussions for reporting errors was suspected as the cause of such low reporting rates [48]. Interestingly, a study previously published by us revealed that final-year medical students had this same reluctance to report adverse events.

Medical students are often overlooked as valuable assets in ensuring PS during clinical rotations. However, assigning them cases based on real personal experiences fosters experiential learning and enhances their understanding and application of theoretical knowledge [49,50]. Although medical students are an underutilized resource for detecting, reporting, and preventing errors [51], completing reporting exercises, like the one presented during our course, could help them overcome barriers to reporting events.

Assessing the impact of medical training courses and programs on PS in relation to patient outcomes is crucial for medical education. However, establishing the true cause–effect is extremely complex [9], as improvement or deterioration in safety and quality in healthcare systems is dependent on many context-specific factors [52]. This challenge is further compounded by ongoing debates about the best methods for measuring patient outcomes influenced by diverse educational policies and study design limitations, complicating the creation of reliable and reproducible studies [53]. We measured the impact of PS and HQI education using test and activities’ scores. Although test scores do not always directly translate to clinical practice, they provide a controlled and measurable variable for evaluating the impact of these educational programs.

## 5. Conclusions

In conclusion, this study highlights the effectiveness of using a combination of innovative teaching methods, technology, and engaging activities to enhance student understanding of PS and HQI. The advantages of our educational intervention include the use of real-life videos, interactive activities, and active learning exercises, which were effective in fostering practical skills and empathy and enhancing student engagement, thus demonstrating promise in improving the quality of care and patient safety outcomes. However, challenges remain in translating theoretical knowledge to practical applications within a tight timeframe, and there were technical difficulties with some activities, such as the augmented reality patient safety simulator. Further longitudinal studies and in-depth investigations of adverse events, along with former students’ experiences, are warranted to explore the long-term impact of these educational strategies in improving primary care quality.

## Figures and Tables

**Figure 1 healthcare-12-01617-f001:**
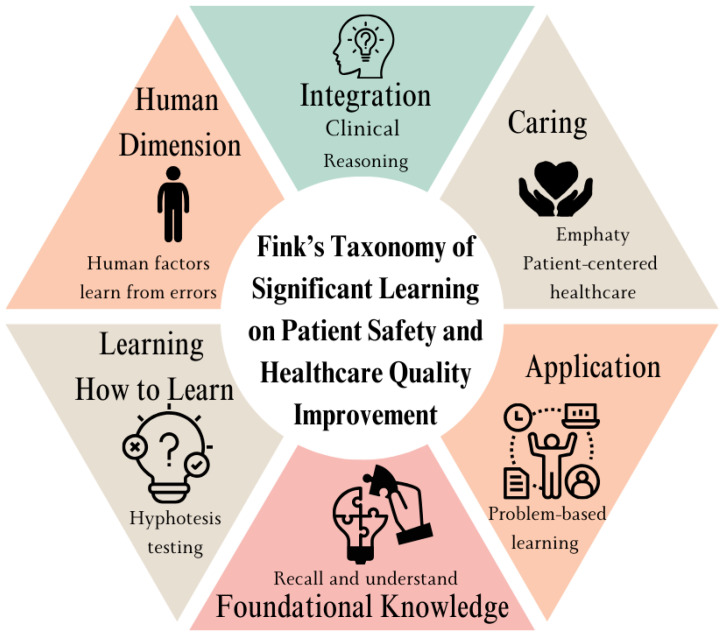
Fink’s significant learning categories as a framework for designing learning activities in teaching patient safety (PS) and healthcare quality improvement (HQI)**.** This figure was created by the authors based on L. Dee Fink’s work [25].

**Figure 2 healthcare-12-01617-f002:**
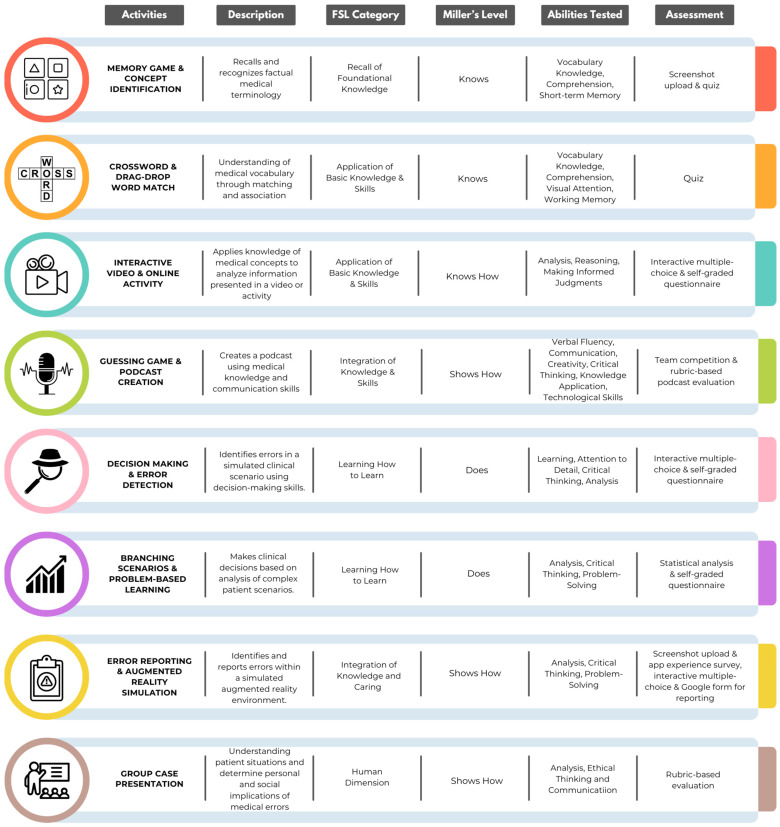
Description of the seven types of reinforcement activities created and used in the course. Legend: FSL = Fink’s significant learning category.

**Figure 3 healthcare-12-01617-f003:**
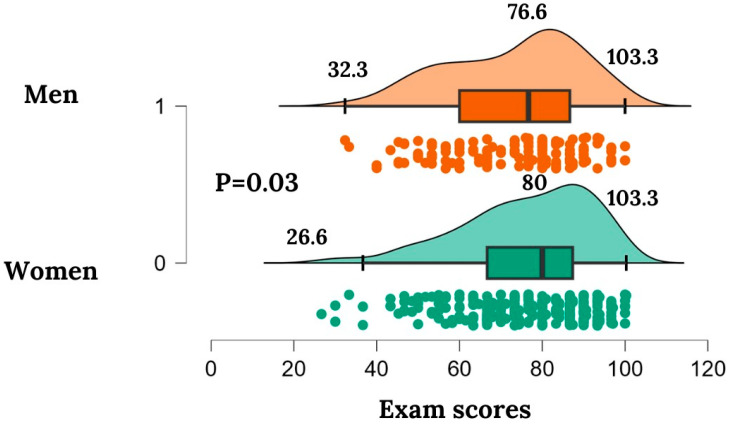
Students’ final exam test scores comparison between men and women including the median, minimum, and maximum values of each group.

**Figure 4 healthcare-12-01617-f004:**
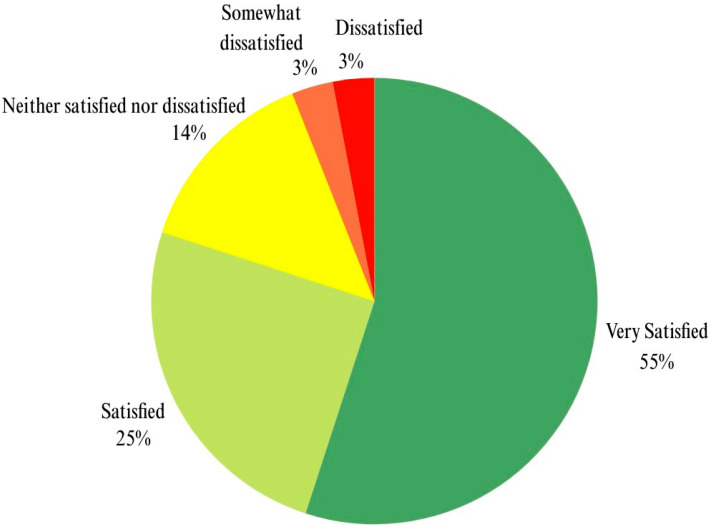
Results of the post-course questionnaire on satisfaction with the course activities.

**Table 1 healthcare-12-01617-t001:** Minimum, maximum, and mean scores obtained by students in each activity.

Day	Activities	Mean	Min	Max	NC	N
1	A. Memory game	4.753	1	5	13	381
	B. Concept identification	4.819	2	5	18	376
2	A. Crossword	4.717	1.5	5	7	387
	B. Drag–drop word match	4.969	3	5	6	388
3	A. Interactive video	7.82	5	8	5	389
	B. Online activity	1.979	0.75	2	3	391
4	A. Guessing game	0.994	0.5	1	7	387
	B. Podcast creation	8.992	6	9	1	393
5	A. Interactive video	4.988	2	5	9	385
	B. Online activity	4.738	1	5	5	389
6	A. Decision-making	2.849	0.5	3	2	392
	B. Error detection	6.847	0.7	7	4	390
7	A. Branching scenarios	4.936	2	5	3	391
	B. Problem-based learning	4.881	2	5	0	394
8	A. Error reporting	4.931	1	5	4	390
	B. Augmented reality simulation	5.39	1.4	6	4	390
9	A. Group case presentation	5.635	3	6	7	387
	B. Group case presentation	5.363	2	6	8	386

Legend: Min: minimum; Max: maximum; NC: not completed; N: completed assignments.

**Table 2 healthcare-12-01617-t002:** Patient safety incident report results (Activity 8A).

Question	Answers	
	Correct	Incorrect
Was there harm to the patient?	99.7% Yes	0.3% No
Incident related to:	79.6% Severe adverse event	20.4% Other
Main International Patient Safety Goal (IPSG) to Prevent Incident:	97.4% Goal 6	2.5% Other
Hospital stay was prolonged due to incident?	94.1% Yes	5.9% No
Contributing factor(s) to the Incident:	17.8% Correct	82.2% Incorrect
What could have happened (potential consequences)?	98.6% Correct	1.4% Incorrect

**Table 3 healthcare-12-01617-t003:** Learner feedback on the AR patient safety simulator (Activity 8B).

Question	Answer	
Did you have any difficulties doing the activity with the simulator?		
Group	Women	Men
Yes	160 (65%)	90 (61%)
No	86 (35%)	58 (39%)
Rate the experience of using the simulator for patient safety		
Group	Women	Men
Very dissatisfied	15 (6.1%)	8 (5.4%)
Dissatisfied	21 (8.5%)	10 (6.7%)
Neutral	44 (17.8%)	25 (16.8%)
Satisfied	62 (25.2%)	41 (27.7%)
Very satisfied	104 (42.2%)	64 (43.2%)
How applicable do you think what you learned with the simulator is to real hospital experience?		
Group	Women	Men
N	246	148
Median	8	9
Mean	7.9	8
Std. Deviation	2.4	2.2
Minimum	0	0
Maximum	10	10
How difficult did you find it to detect patient risks using the simulation app?		
Group	Women	Men
N	246	148
Median	6	6
Mean	5.66	5.9
Std. Deviation	2.6	2.6
Minimum	0	0
Maximum	10	10

## Data Availability

The data presented in this study are available from the corresponding author upon request.

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
