# Peer review of "Student Engagement in Patient Safety and Healthcare Quality Improvement: A Brief Educational Approach"

_healthcare, 2024, doi:10.3390/healthcare12161617_

Round 1
Reviewer 1 Report
Comments and Suggestions for Authors
The article presents an interesting proposal to modify the method of educating medical students. Its content is adapted to the topic of the special issue.
The Introduction chapter should include information about the place of the topic PS and HQI in the education programs for the field of study in question. What is the current research on this topic?
- Fig.1 Who is the author of the graphic? Does the process shown in the diagram come from previous research/articles? There should be a footnote - similarly, subsequent illustrations.
- The term “medical students” means future doctors or candidates for nurses?
- The description of the methodology lacks a schedule for the research performed. Demographic characteristics related to the respondent's occupation should be included.
- The "conclusions" chapter should be expanded to include information on the advantages and possible disadvantages of the proposed educational method.
- The article requires re-editing, regarding the list of references.
Comments on the Quality of English LanguageMinor editing of English language required
Author Response
Thank you very much for taking the time to review this manuscript. Please find the detailed responses below and the corresponding revisions marked in red in the re-submitted files.
Comment 1: The Introduction chapter should include information about the place of the topic PS and HQI in the education programs for the field of study in question. What is the current research on this topic?
Response 1: We have expanded the introduction and added a paragraph about the current role of PS and HQI in the medical curriculum, as well as multiple examples of other approaches taken for PS and HQI educational interventions (lines 43-55).
Comment 2: Fig.1 Who is the author of the graphic? Does the process shown in the diagram come from previous research/articles? There should be a footnote - similarly, subsequent illustrations.
Response 2: We have added a footnote stating that the figure was created by the authors based on L. Dee Fink’s research on creating significant learning experiences, which we have cited in the footnote.
Comment 3: The term “medical students” means future doctors or candidates for nurses?
Response 3: In the context of this study, the term “medical students” is used to refer to future doctors. The findings could also apply to other healthcare professionals.
Comment 4: The description of the methodology lacks a schedule for the research performed.
Response 4: We have added a schedule for the activities performed for this research in segment 2.4 of the methods section (lines 186-190).
Comment 5: Demographic characteristics related to the respondent's occupation should be included.
Response 5: All respondents had the occupation of full-time university students. If some of them had another occupation outside of school, we didn’t think it relevant to our research and therefore that data was not collected.
Comment 6: The "conclusions" chapter should be expanded to include information on the advantages and possible disadvantages of the proposed educational method.
Response 6: We have expanded the conclusion section to include a discussion of the perceived advantages and disadvantages of our educational intervention. We hope these revisions meet your expectations and improve the overall quality of the manuscript.
Comment 7: The article requires re-editing, regarding the list of references.
Response 7: We have used a reference manager software (EndNote) that automatically re-edits references to the journal’s style.
Response to comments of Quality of English Language
We have utilized an AI engine to correct and enhance the quality of the English language in the manuscript.
Reviewer 2 Report
Comments and Suggestions for Authors
This study aimed to demonstrate that innovative teaching methods and technologies can help medical students better understand and apply patient safety and healthcare quality principles and contribute to improving the quality of care while achieving optimal patient safety, which remains a challenging task in healthcare. This study is attractive in terms of its subject, but for it to be published in the Healthcare journal, the authors must review some of the deficiencies stated below.
1. The abstract of the study can be expressed more briefly. Unnecessary explanations should be included in the abstract.
2. If the abstract of the study focuses on PS and HQI, HQI should be included in the keywords.
3. Line 56-59: The statements in the introduction of the study must be scientifically supported. Otherwise, this is understood only as the opinion of the authors.
4. Line 59-60: The sentence "Miller's Pyramid of Clinical Competence is a framework for assessing clinical skills in medical education" should be supported by a citation.
5. In this study, the authors propose a new educational approach using Fink's Taxonomy to bridge the gap between theoretical knowledge and clinical practice in HQI and PS. Authors should write why they adopted this approach. What is the purpose of using this method?
6. Figure 2, included in the methodology, could be more visually pleasing.
7. the mathematical structures of the methods preferred for the study or the statistical methods shared in the result section should be mentioned in the methodology. The methodology part of the study should be expanded.
8. Figure 3 should be reviewed.
9. Figure 4 should be reviewed.
10. The authors have structurally handled the study well. However, the methodology and conclusion sections of the study need to be revised. These sections should be reviewed.
Comments on the Quality of English LanguageMinor editing of English language required
Author Response
Thank you very much for taking the time to review this manuscript. Please find the detailed responses below and the corresponding revisions marked in red in the re-submitted files.
Comment 1: The abstract of the study can be expressed more briefly. Unnecessary explanations should be included in the abstract.
Response 1: We believe that the abstract reflects the main and most important ideas of the study and is within the limit of words provided. However, we have made minor corrections to the abstract to avoid redundancy. Could you specify what explanations or sentences did you find unnecessary?
Comment 2: If the abstract of the study focuses on PS and HQI, HQI should be included in the keywords.
Response 2: We have added HQI as a keyword.
Comment 3: Line 56-59: The statements in the introduction of the study must be scientifically supported. Otherwise, this is understood only as the opinion of the authors.
Response 3: Reference [20] has been included to support the statement.
Comment 4: Line 59-60: The sentence "Miller's Pyramid of Clinical Competence is a framework for assessing clinical skills in medical education" should be supported by a citation.
Response 4: The citation supporting this statement [22] has been added at the end of the sentence.
Comment 5: In this study, the authors propose a new educational approach using Fink's Taxonomy to bridge the gap between theoretical knowledge and clinical practice in HQI and PS. Authors should write why they adopted this approach. What is the purpose of using this method?
Response 5: The advantages and our reasoning for using Fink’s Taxonomy in the creation of our course are mentioned in the introduction section in lines 82-87 and 108-111.
Comment 6: Figure 2, included in the methodology, could be more visually pleasing.
Response 6: Figure 2 has been reviewed and modified to be more visually pleasing.
Comment 7: The mathematical structures of the methods preferred for the study or the statistical methods shared in the result section should be mentioned in the methodology. The methodology part of the study should be expanded.
Response 7: We agree with this comment, the methods section of the article has been revised with multiple modifications. We have expanded sections 2.3 and 2.4 to provide further clarification on the course details and study development. Additionally, we have added section 2.6, which outlines the specifics of the statistical data analysis.
Comment 8: Figure 3 should be reviewed.
Response 8: Figure 3 has been revised and we have added the median, maximum and minimum values so that the data represented is better understood.
Comment 9: Figure 4 should be reviewed.
Response 9: We have reviewed Figure 4 and we believe that it presents relevant data in a clear manner.
Comment 10: The authors have structurally handled the study well. However, the methodology and conclusion sections of the study need to be revised. These sections should be reviewed.
Response 10: Thank you for your constructive feedback. We appreciate your positive comments about the overall structure of the study. In response to your suggestion, we have thoroughly revised the methodology and conclusion sections to ensure clarity and completeness. We have provided additional details and addressed any ambiguities to enhance the robustness of these sections.
Response to comments of Quality of English Language
We have utilized an AI engine to correct and enhance the quality of the English language in the manuscript.
Reviewer 3 Report
Comments and Suggestions for Authors
Dear authors,
The manuscript's topic is very current and significant in terms of a very challenging issue in the healthcare system: achieving optimal patient safety. The study makes a special contribution because it indicates the application of innovative teaching methods, which is always a great challenge for both the teacher and the students. The manuscript also provides significant empirical data indicating the positive effects of innovative teaching methods and technology on students' understanding of patient safety and improving healthcare quality.
I would like to offer some suggestions to improve the manuscript:
- Introduction: I suggest that the facts in lines 34-37 be supported by references from the literature. In line 44, you stated that there are different approaches to the PS and HQI courses. However, it would be helpful to cite other authors' approaches to give readers a better insight into the problem the manuscript deals with. It would also be helpful to provide examples of studies and courses conducted with medical students in which the authors relied on Miller's Pyramid of Clinical Competencies and Fink's Taxonomy (lines 59-76).
Figure 1 clearly provides insight into Fink's Significant Learning Categories as a framework for designing learning activities in teaching PS and HQI. However, information on whether the authors independently created or adapted it (mention the reference/es) is missing.
Line 81: Is the course program on PS and HQI for fourth-year medical students mandatory or optional?
- Materials and Methods: in line 101 states that the program is intended to be completed within two weeks. Does the entire course take place during this period? If it is part of the course, in which part of the course was the program implemented and what determined the author's decision? Are classes from other courses held at the same time?
Modified educational activities, seven reinforcement types. are clearly described in the text and graphically in Figure 2.
Line 147-153. It is necessary to state what the standardized test is. Who created the question bank? Where was the exam taken in a university environment or somewhere else? Did all students take the exam at the same time? Is the total score calculated, and how? Were there prescribed threshold values for passing/failing the exam, and what were the thresholds?
In lines 155-160, It is necessary to specify the criteria for the inclusion of students in the study. Are these all students enrolled in the course? Did they have the option to decline participation? In that case, how did they attend the classes? Did that reflect on their success in the courses? What is the specific contribution of the four professors to this study?
In line 183 it would be useful to specify the range of answers? What is the range of the total score?
Although the instructions for authors and the journal template do not oblige authors to create a data analysis/statistical data processing section, they would significantly contribute to a more adequate presentation of work methods. I suggest the authors to create it.
- Results: Table 1 lacks acronyms. In line 204, I suggest the authors list the other values ​​for the Mann-Whitney test (Md, U, z) in addition to the p-value. Certainly, information on the impact's effect size would strengthen the significance of the obtained results.
In Figure 3, the axes must be clearly marked. The figure's name does not include a student.
- Discussion: The authors draw attention to many important questions raised by their research in correlation with the results of other authors. Furthermore, they clearly point out the advantages of their studies, suggest potential shortcomings, and discuss aspects of education that would further encourage students to recognize and solve medical errors.
I hope you find my comments helpful.
Author Response
Thank you very much for taking the time to review this manuscript. Please find the detailed responses below and the corresponding revisions marked in red in the re-submitted files.
Comment 1: Introduction: I suggest that the facts in lines 34-37 be supported by references from the literature.
Response 1: The reference [2] to support this fact has been added.
Commet 2: In line 44, you stated that there are different approaches to the PS and HQI courses. However, it would be helpful to cite other authors' approaches to give readers a better insight into the problem the manuscript deals with. It would also be helpful to provide examples of studies and courses conducted with medical students in which the authors relied on Miller's Pyramid of Clinical Competencies and Fink's Taxonomy (lines 59-76).
Response 2: We have expanded the introduction and added a paragraph about the current role of PS and HQI in the medical curriculum as well as multiple examples of different approaches taken for PS and HQI educational interventions with their corresponding reference (lines 43-55 and 66-87).
Comment 3: Figure 1 clearly provides insight into Fink's Significant Learning Categories as a framework for designing learning activities in teaching PS and HQI. However, information on whether the authors independently created or adapted it (mention the reference/es) is missing.
Response 3: We have added a footnote stating that the figure was created by the authors based on L. Dee Fink’s research on creating significant learning experiences, which we have cited in the footnote.
Comment 4: Line 81: Is the course program on PS and HQI for fourth-year medical students mandatory or optional?
Response 4: The course is mandatory in the university’s curriculum; we have specified this in the manuscript.
Comment 5: Materials and Methods: in line 101 states that the program is intended to be completed within two weeks. Does the entire course take place during this period? If it is part of the course, in which part of the course was the program implemented and what determined the author's decision? Are classes from other courses held at the same time?
Response 5: We initially used "program" and "course" interchangeably alluding to the same thing, which caused confusion. We have now replaced "program" with “course” for consistency. The entire course spans two weeks, and the syllabus was specifically designed to fit within this timeframe (line 120-121). During the course duration students also had classes from a course on geriatrics, but it wasn’t related to our syllabus.
Comment 6: Modified educational activities, seven reinforcement types. are clearly described in the text and graphically in Figure 2.
Response 6: We have modified the design of Figure 2 per one reviewer’s request, but the content remains the same.
Comment 7: Line 147-153. It is necessary to state what the standardized test is. Who created the question bank? Where was the exam taken in a university environment or somewhere else? Did all students take the exam at the same time? Is the total score calculated, and how? Were there prescribed threshold values for passing/failing the exam, and what were the thresholds?
Response 7: The question bank was created by the course professors and students of each cycle of the course took the exam at the same time on campus. To answer all your questions and clarify what the final exam entailed, we have expanded on the final exam’s characteristics, how it was applied and graded in segment 2.3 of the methods section.
Comment 8: In lines 155-160, It is necessary to specify the criteria for the inclusion of students in the study. Are these all students enrolled in the course? Did they have the option to decline participation? In that case, how did they attend the classes? Did that reflect on their success in the courses? What is the specific contribution of the four professors to this study?
Response 8: Students enrolled in the course in the 2023-2024 academic year were invited verbally and through the questionnaire to participate in this educational research project on a voluntary basis. To obtain honest and unbiased responses, we ensured students understood that their participation would not affect their grades and that their responses would be anonymous. Registration numbers were collected solely to avoid duplicate entries and would be kept confidential. The other four professors contributed by teaching the classes and distributing the questionnaire, they weren’t involved in the instructional design of the course. All this has been clarified in segment 2.4.
Comment 9: In line 183 it would be useful to specify the range of answers? What is the range of the total score?
Response 9: Students rated the activities based on how much they felt it helped them prepare in the for the exam on a score of 0 to 100. The activity with the highest score was activity 8A with a score of 85 out of 100 (line 264).
Comment 10: Although the instructions for authors and the journal template do not oblige authors to create a data analysis/statistical data processing section, they would significantly contribute to a more adequate presentation of work methods. I suggest the authors to create it.
Response 10: We agree with this feedback and have added section 2.6 in the methodology, which outlines the specifics of the statistical data analysis.
Comment 11: Results: Table 1 lacks acronyms.
Response 11: We have added a legend that contains the meaning of the abbreviations and acronyms used on the table so that the information presented is clear.
Comment 12: In line 204, I suggest the authors list the other values ​​for the Mann-Whitney test (Md, U, z) in addition to the p-value. Certainly, information on the impact's effect size would strengthen the significance of the obtained results.
Response 12: The other values of the Mann-Whitney test have been added.
Comment 13: In Figure 3, the axes must be clearly marked. The figure's name does not include a student.
Response 13: Figure 3 has been revised and we have added the median, maximum and minimum values so that the data represented is better understood. The name of the figure has also been changed to “Students’ final exam test scores comparison between men and women including median, minimum and maximum values of each group”.
Comment 14: Discussion: The authors draw attention to many important questions raised by their research in correlation with the results of other authors. Furthermore, they clearly point out the advantages of their studies, suggest potential shortcomings, and discuss aspects of education that would further encourage students to recognize and solve medical errors.
Response 14: Thank you for your constructive feedback. We appreciate your positive comments and suggestions about the study. We hope these revisions meet your expectations and improve the overall quality of the manuscript.